# A Tetra-Panel of Serum Circulating miRNAs for the Diagnosis of the Four Most Prevalent Tumor Types

**DOI:** 10.3390/ijms21082783

**Published:** 2020-04-16

**Authors:** Belén Pastor-Navarro, María García-Flores, Antonio Fernández-Serra, Salvador Blanch-Tormo, Fernando Martínez de Juan, Carmen Martínez-Lapiedra, Fernanda Maia de Alcantara, Juan Carlos Peñalver, José Cervera-Deval, José Rubio-Briones, Jaime García-Rupérez, José Antonio López-Guerrero

**Affiliations:** 1Laboratory of Molecular Biology, Fundación Instituto Valenciano de Oncología, 46009 Valencia, Spain; bpastor@fivo.org (B.P.-N.); mgarciaf@fivo.org (M.G.-F.); afernandez@fivo.org (A.F.-S.); 2IVO-CIPF Joint Research Unit of Cancer, Príncipe Felipe Research Center (CIPF), 46012 Valencia, Spain; 3Department of Medical Oncology, Fundación Instituto Valenciano de Oncología, 46009 Valencia, Spain; sblanch@fivo.org; 4Unit of Gastroenterology, Fundación Instituto Valenciano de Oncología, 46009 Valencia, Spain; fmartinezj@fivo.org (F.M.d.J.); mmartinezl@fivo.org (C.M.-L.); fmalcantara@fivo.org (F.M.d.A.); 5Department of Thoracic Surgery, Fundación Instituto Valenciano de Oncología, 46009 Valencia, Spain; jcpenalver@fivo.org; 6Department of Radiology, Fundación Instituto Valenciano de Oncología, 46009 Valencia, Spain; jcervera@fivo.org; 7Department of Urology, Fundación Instituto Valenciano de Oncología, 46009 Valencia, Spain; jrubio@fivo.org; 8Nanophotonics Technology Center, Universitat Politècnica de València, 46022 Valencia, Spain; jaigarru@upvnet.upv.es; 9Department of Pathology, School of Medicine, Catholic University of Valencia ‘San Vicente Mártir’, 46001 Valencia, Spain

**Keywords:** Early diagnosis, circulating microRNAs, lung cancer, breast cancer, colorectal cancer, prostate cancer

## Abstract

The purpose of this study is to clinically validate a series of circulating miRNAs that distinguish between the 4 most prevalent tumor types (lung cancer (LC); breast cancer (BC); colorectal cancer (CRC); and prostate cancer (PCa)) and healthy donors (HDs). A total of 18 miRNAs and 3 housekeeping miRNA genes were evaluated by qRT-PCR on RNA extracted from serum of cancer patients, 44 LC, 45 BC, 27 CRC, and 40 PCa, and on 45 HDs. The cancer detection performance of the miRNA expression levels was evaluated by studying the area under the curve (AUC) of receiver operating characteristic (ROC) curves at univariate and multivariate levels. miR-21 was significantly overexpressed in all cancer types compared with HDs, with accuracy of 67.5% (*p* = 0.001) for all 4 tumor types and of 80.8% (*p* < 0.0001) when PCa cases were removed from the analysis. For each tumor type, a panel of miRNAs was defined that provided cancer-detection accuracies of 91%, 94%, 89%, and 77%, respectively. In conclusion, we have described a series of circulating miRNAs that define different tumor types with a very high diagnostic performance. These panels of miRNAs would constitute the basis of different approaches of cancer-detection systems for which clinical utility should be validated in prospective cohorts.

## 1. Introduction

Nowadays, cancer is expected to rank as the leading cause of death in every country of the Western world in the 21st century. The Global Cancer Observatory (GLOBOCAN) 2018 estimated that there would be 18.1 million new cancer cases and 9.6 million cancer deaths worldwide in 2018. Globally, the incidence rate for all cancers combined was about 20% higher in men than in women, with the incidence rates varying across regions in both males and females, indicating that 1 in 8 men and 1 in 10 women will develop cancer along their life. For both sexes combined, lung cancer (LC) is the most commonly diagnosed cancer (11.6% of the total cases), followed by female breast cancer (BC) (11.6%), colorectal cancer (CRC) (10.2%), and prostate cancer (PCa) (7.1%) [1].

Despite the high mortality of this disease, it has been demonstrated that the chance of curing cancer is very high when identified early. In this sense, cancer screening programs (CSPs) play a critical role in identifying cancer before symptoms appear, and their impact on cancer-specific and overall survival has been well documented [2]. Many of these CSPs may involve analytical procedures such as blood or urine tests, other tests, or medical imaging that identifies individuals with a high probability of having cancer and whose diagnosis should be confirmed by means of a histopathological examination.

Many of these analytical tests are based on the identification of blood-based tumor biomarkers which constitute a readily available, inexpensive, and minimally invasively tool which also allows for repeated sampling [3].

Particularly, microRNAs (miRNAs) are 19–25 nucleotide noncoding RNA molecules that regulate a variety of cellular processes including cell differentiation, cell cycle progression, and apoptosis [4]. Some studies have reported that some circulating miRNAs in serum and plasma can be used as noninvasive biomarkers for defining different stages of disease, including cancer [5].

To take advantage of miRNAs as biomarkers for the early identification of cancer, the H2020 European project SAPHELY (Self-amplified photonic biosensing platform for microRNA-based early diagnosis of diseases) (https://saphely.eu/) is focused on developing a nanophotonic-based handheld point of care nanophotonic device for detecting specific circulating miRNAs for the four major cancer types: LC, BC, CRC, and PCa.

Hence, the purpose of this study is to clinically validate a series of circulating miRNAs that distinguish between the different tumor types and controls and that could constitute the basis of the SAPHELY detection system.

## 2. Results

### 2.1. Average of Expression Levels of Housekeeping Genes U6snRNA, miRNA-16, and miRNA-1228 as Reference in the Serum of HDs and Cancer Patients

To quantify the expression of the selected circulating miRNAs, it was necessary to select a reference housekeeping miRNA or combination of miRNAs [6] with stable expression levels in the serum from both healthy donors (HDs) and cancer patients. Three housekeeping miRNAs were evaluated: U6snRNA, miR-16, and miR-1228 [7,8,9].

The average Ct value of each housekeeping miRNA or the mean Ct values of the combination of 2 or 3 of these miRNAs were evaluated to select the best combination of miRNAs that did not show differences between the groups under study. A nonparametric test for independent samples based on the median of the Ct values was used to demonstrate that no significant differences between groups were observed (Figure 1). This analysis shows that the Ct average of the three housekeeping genes constituted the best option for normalizing the circulating miRNAs expression levels.

### 2.2. Analysis of miR-21

miR-21 was significantly overexpressed in all cancer types compared with HDs (Figure 2), suggesting that only this miRNA might be indicative for the presence of cancer. In order to analyze the diagnostic performance of circulating miRNA-21, a ROC curve was calculated, obtaining an AUC of 0.675 (95% CI: 0.578–0.736; *p* = 0.001) (Figure 3A). Interestingly, when PCa patients were removed from the analysis (because the dispersion of miR-21 expression in this cohort was very high), the diagnostic performance of miR-21 increased in almost 15%: AUC of 0.808 (95% CI: 0.738–0.878; *p* < 0.0001) (Figure 3B).

### 2.3. Definition of Circulating miRNA Panels to Identify Cancer Patients

As for miR-21, the diagnostic performance of the other circulating miRNAs was evaluated with ROC curves for each tumor type. The results are summarized in Table 1.

The analysis was performed separately for each tumor type considering those tumor-specific miRNAs (Appendix A). For each miRNA, the AUC was calculated, indicating that not all selected miRNAs could be useful for distinguishing between cancer patients and HDs (Table 1). According to this analysis, the following set of miRNAs significantly identified specific cancer patients: LC (miR-21, miR-429, miR-200b, and miR-125b) (Figure 4A); PCa (miR-21, miR-141, miR-375, miR-125b, and miR-182) (Figure 4B); CRC (miR-21, miR-29a, miR-210, miR-200c, and miR-155) (Figure 4C); and BC (miR-21 and miR-205) (Figure 4D).

### 2.4. Multivariate Analysis

In order to adjust the weight of each miRNAs in detecting cancer, a Fisher linear discriminant analysis was performed for each tumor type using only those miRNAs that significantly identified cancer patients (those with an AUC > 0.5 and *p*-value < 0.05) (Table 1). ROC curves for these models showed that the combinations of the expression levels of circulating miRNAs improved the discriminant diagnostic capability of individual miRNAs (Table 2 and Figure 5). Therefore, the diagnostic accuracy in identifying cancer patients with these models are 77%, 89%, 91%, and 94% for BC, CRC, LC, and PCa, respectively.

In order to validate the power of the statistical model, a cross-validation consisting of 1000 iterations with replacement was performed for each tumor type. The strategy was to repeat cross-validation at intervals of 10 samples (i.e., 30, 40, 50, …, *n* with *n* being the sample size of each of the four datasets). Every subset is randomly resampled in each iteration up to 1000. These subsamples show that stabilization of the AUC along with standard deviation shrinkage is achieved before reaching the sample size studied in this work, showing an asymptotic pattern and suggesting that an increase of the number of cases for each tumor type would not significantly change the observed AUC (Appendix A).

### 2.5. Analysis of Variation of the Tumor-Selected Circulating-miRNA Depending on Tumor Stage

In order to demonstrate if the tumor-selected circulating-miRNAs are associated to tumor stages, a nonparametric test for independent samples was performed. The results showed a significant increase (*p* ≤ 0.001) in the probability of being classified as cancer cases as tumor stage rises, corroborating the performance of miRNAs selected as diagnostic and prognostic tools (Appendix A).

Moreover, a nonparametric test was also performed in PCa samples to analyze the probability of being classify as cancer cases based on the variation of the PCa-selected circulating-miRNA depending on Gleason Score (GS) and prostate specific antigen (PSA) values. In this case, despite the fact that the median values of the probabilities are higher in GS, ≥7, and serum PSA, ≥4.33 ng/mL, the differences were not statistically significant (*p* > 0.05) (Appendix A), mainly because of the disbalancing of the groups analyzed, with only 15% of cases belonging to advanced disease.

## 3. Discussion

Cancer represents the set of diseases with more incidences and mortality in Western countries, and its numbers are increasing every year with the consequent social and economic impacts [1]. For that reason, special efforts have been allocated to the design of efficient CSP for specific tumor types that impact overall survival [10,11,12,13]. Most of these CSP incorporate screening tests for selecting individuals with risk of having cancer to whom the standard diagnostic procedures are applied to confirm the disease. Unfortunately, most of these tests have demonstrated poor accuracy and efficacy, particularly among the most prevalent cancers [14]. The actual cancer-screening tests for the four most prevalent tumor types (CRC, LC, PCa, and BC) are mainly based on radiological images (Computed Axial Tomography (CAT), mammography, and multiparametric magnetic resonance for LC, BC, and PCa respectively), some biomarkers such as PSA in PCa or fecal occult blood for CRC, and other invasive interventions such as colonoscopy for CRC [11,12,13,14]. These cancer-screening tests are characterized by low sensitivities or specificities (PSA and fecal occult blood tests) or require specialized trained personnel (image related tests) that make them expensive and limit their use to specific populations and/or referral centers [14]. This is the reason by which blood biomarkers have been proposed as effective indicators to distinguish between cancer and normal conditions or among different cancer groups [15].

More than one decade ago, it was reported that miRNAs are also present in blood [16]. Circulating miRNAs were found to be remarkably stable even under conditions as harsh as boiling, low or high pH, long-time storage at room temperature, and multiple freeze-thaw cycles [5]. Furthermore, miRNAs also represent the status of the disease as they are associated with tumor biology and tumor behavior [17,18]. Thus far, distinctive patterns of circulating miRNAs have been found for different tumor types [19] including BC [20], PCa [21], LC [22], and CRC [23].

Taking all these premises into consideration, the H2020-SAPHELY project (https://saphely.eu) intends to break into the field of screening tests for early diagnosis of the four major cancer types, PCa, BC, LC, and CRC, by using a novel ultrahigh sensitivity nanophotonic-based sensing technique for the direct detection of circulating miRNA biomarkers through a combination of molecular beacon probes with an attached high index nanoparticle so that the hybridization events are translated into the displacement of these nanoparticles from the sensor surface [24,25]. The idea of the SAPHELY device because of the final cost of the test and the expected accuracy in cancer detection is that it could be useful in a cancer screening context. For an explanation of how the SAPHELY device would work, please visit the following URL: https://youtu.be/6ZAuSkfJrB8?list=PL8qM5Jl41EI7iW2a57QEyaCvRFHn-ix95.

With this study, we have evaluated, from the clinical point of view, a series of miRNAs that were already defined in independent studies (Appendix A) and that, to some extent, were associated with diagnosis or tumor progression of patients with at least one of the four most prevalent types of tumors (Appendix A). For LC and CRC, two sets of 7 miRNAs each were evaluated; 5 miRNAs were evaluated for PCa; and 3 were evaluated for BC. The expression levels of these miRNAs were analyzed by qRT-PCR and their diagnostic capacity by ROC curves using serum samples from cancer patients and HDs in order to define those miRNAs that finally would be incorporated within the SAPHELY detection system. Based on our results, only 12 miRNAs (miR-21, miR-429, miR-200b, miR-125b, miR-141, miR-375, miR-182, miR-29a, miR-210, miR-200c, miR-155, and miR-205) and three housekeeping genes (U6snRNA, miR-16, and miR-1228) would constitute the basis of the detection system of the SAPHELY device. Combinations of the expression levels of these miRNAs provides diagnosis accuracies of 77%, 89%, 91%, and 94% for BC, CRC, LC, and PCa, respectively (Figure 5 and Table 2).

Interestingly, of all validated miRNAs, miR-21 was overexpressed in all four cancer types, particularly in BC, CRC, and LC. In fact, the diagnostic accuracy for cancer of miR-21 was 67.5% when considering the four tumor types and it increased up to 80.8% when PCa patients were released from the analysis. Overall, miR-21 is considered to be a typical “onco-miR”, which acts by inhibiting the expression of phosphatases, which limit the activity of signaling pathways such as AKT and MAPK. miR-21 is associated with a wide variety of cancers including that of breast [26], ovaries [27], colorectal [28], lung, or prostate [29], among others. More recently, a 2014 meta-analysis of 36 studies evaluated circulating miR-21 as a biomarker of various carcinomas, finding that it has potential as a tool for early diagnosis [30], findings which are in accordance to those herein reported.

The LC miRNA panel is represented by miR-21, miR-429, miR-200b, and miR-125b; except for miR-21, the rest of miRNAs were discovered and validated in a previous study in which we collaborated with Reference [31]. This work defined a panel of 6 miRNAs with a diagnostic accuracy for LC of 89%, which is very close to the 91% described in our study.

We have defined BC detection with miR-21 and miR-205, achieving a 77% diagnostic precision. With this study, we have confirmed the diagnostic value of miR-205 defined by other authors that demonstrated the overexpression of this miRNA in serum of BC patients [32,33].

For PCa, the diagnostic accuracy defined by the miRNA panel (miR-21, miR-141, miR-375, miR-125b, and miR-182) was 94%, the highest of our series and similar to other clinico-molecular multi-biomarker panels focused on detection of high-grade PCa [34]. Whereas circulating miR-141, miR-375, and miR-125b were already described by other authors as potential diagnostic biomarkers [35,36], we have introduced miR-182 as a novel promising circulating miRNA. Overexpression of this miRNA was reported by our group as the strongest prognostic miRNA in PCa, and their expression was directly proportional to tumor stage and Gleason score [37]. Hence, with this study, we have demonstrated that circulating miR-182 (AUC: 0.895; 95% CI: 0.824–0.967; *p* < 0.0001) significantly increases the performance of the PCa miRNA panel.

Finally, CRC miRNA panel (miR-21, miR-29a, miR-210, miR-200c, and miR-155) achieves an 89% diagnostic performance. These miRNAs have been already described by other authors [4,38,39], and our study validates in an independent cohort of cases their findings.

Additionally, due the role of miRNA as key regulators of virtually every biological process, it is interesting to study the impact of miRNA in phenotypes by studying the role of its effector mRNAs. In this sense, there is a great corpus of evidence from both functional studies and computational predictions. This data is organized in public databases on the Internet along which one of the most important is miRBase [40].

An in silico analysis reveals a dense network of gene effectors of the miRNA in every one of the four tumor types studied. Gene Set Enrichment Analysis (GSEA) highlighted the three top overrepresented pathways. In all four datasets, the main Kyoto Encyclopedia of Genes and Genomes (KEGG) category is pathways related to cancer. In LC, this category is accompanied by ErbB signaling and PCa pathways, whereas PCa showed altered pancreatic and PCa pathways. CRC and finally the BC series have overrepresented pancreatic and PCa pathways. There is difficulty drawing direct conclusions of such a dense network of genes; thus, the network was previously pruned according the degree of the nodes in every one of the disease set. The top overrepresented KEGG pathways are closely related with cancer in all the cases with interesting potential mRNA biomarkers which deserve to be explored (Appendix A).

Moreover, we have demonstrated that the tumor-selected circulating-miRNA are associated to different tumor stages but not to GS and serum PSA values in the case of PCa. This observation gives robust evidence of circulating-miRNA utility as a diagnostic and prognostic tool in these four cancer types.

However, this present study has some limitations. In the first place, we only have analyzed some miRNAs considered tumor specific for each cancer type, but there are more miRNAs that have not been investigated that could be relevant for cancer diagnosis. Moreover, we need an amplification step in order to increase the miRNA concentration to being able to detect them, but the SAPHELY device is not be able to perform this step, a point that could be crucial for the success of the device. Finally, we have defined a panel for each cancer type, but they are not always represented, with all altered miRNAs in all cases hindering the disease identification.

Because of these limitations, this present study must be continued by analyzing more miRNAs that can be completely tumor-specific and by improving the detection system to offset the low concentration of miRNAs in bloodstream.

## 4. Patients and Methods

### 4.1. Ethics Statement

This study was approved by the Clinical Research Ethics Committee (CREC) of the Fundación Instituto Valenciano de Oncología of Valencia (FIVO) at the meeting celebrated on 30 December 2014. Written informed consent was obtained from each participant prior to sample collection.

### 4.2. Patients’ Characteristics

Patients with newly diagnosed LC, BC, CRC, and PCa were considered eligible for this study. Within the period from October 2011 to March 2017, a total of 156 cancer patients (44 LC, 40 PCa, 27 CRC, and 45 BC) was finally selected. Patients’ clinical and histopathological information are summarized in Table 3. Serum samples from 45 HDs with no history of cancer or other chronical disease prior the blood collection were also collected. These samples were included as a control group for the miRNA analyses.

### 4.3. MiRNAs Selection for Each Tumor Type

In view of the role of miRNAs as diagnostic biomarkers, a literature search was performed to identify those miRNA candidates to be implemented as biomarkers. A list of potential miRNAs was selected for each tumor type by applying a decision-making algorithm using PubMed as source of information. The search criteria used were (“microRNA” OR “miRNA” OR “miRNAs”) AND (“prostate cancer”/“breast cancer”/“lung cancer”/“colorectal cancer”) AND (“serum” OR “plasma”) AND (“diagnosis”). All the papers selected met the following criteria: expression of miRNAs in serum; papers written in English; and clinical setting: diagnosis, including control samples.

The results of the literature search are summarized in Appendix A.

### 4.4. Blood Collection and RNA Extraction

Seven milliliters of peripheral blood from cancer patients and HDs were collected at the time of diagnosis in SARSTEDT Monovette Serum gel tubes (Sarstedt AG & Co, Nümbrecht, Germany) and left to clot at room temperature (RT) for 30 min. The serum was separated by centrifugation for 10 min 3000× *g* at RT and stored at FIVO Biobank at −80 °C in 1-mL aliquots until RNA could be extracted and purified.

Total RNA was isolated using the miRNeasy Serum/Plasma Kit (Qiagen N.V., Hilden, Germany) following the manufacturer’s instructions, starting with 200 µL of serum. The concentration of total RNA in each sample was measured using a NanoDrop 1000 spectrophotometer (Thermo Scientific, Wilmington, DE).

### 4.5. Conversion of total RNA into cDNA

One-hundred and fifty ng of total RNA were reverse-transcribed using the TaqMan MicroRNA Reverse Transcription kit (Applied Biosystems, Foster City, CA) in a total volume reaction of 15 µL containing 2 mM dNTPs, 3.3 U/mL MultiScribe reverse transcriptase, 1× reverse transcription buffer, 0.25 U/mL RNase inhibitor, and 0.02× of each specific miRNA primer (Appendix A), that were pooled from a 5× solution to a working solution of 0.05× (TaqMan MicroRNA Assays, Applied Biosystems) and nuclease-free water. The reaction was performed using the Veriti Thermal Cycler (Applied Biosystems) at 16 °C for 30 min, 42 °C for 30 min, and 85 °C for 5 min.

### 4.6. cDNA Pre-Amplification

Five µL of cDNA was pre-amplified using the TaqMan PreAmp Master Mix (2×) according to the manufacturer’s instructions (Applied Biosystems). For this reaction, 5 µL of cDNA sample was added to a total volume of 25 µL containing 12.5 µL of TaqMan PreAmp Master Mix (2×), 7.25 µL of polled assay mix (0.2×), and nuclease-free water. The reaction was performed using a Veriti Thermal Cycler (Applied Biosystems) during 14 cycles at 95 °C for 10 min, 95 °C for 15 s, and 60 °C for 4 min.

### 4.7. Quantification of Circulating miRNAs by Quantitative Reverse Transcription-Polymerase Chain Reaction (qRT-PCR)

cDNA, previously pre-amplified, was diluted 1:10 in Tris-EDTA (TE) buffer 1× and added to a final qRT-PCR reaction volume of 20 µL, which contained TaqMan MicroRNA assay primers (Applied Biosystems) for each miRNA (Appendix A), TaqMan Gene Expression Master Mix (2×) (Applied Biosystems), and nuclease-free water. The reaction was performed using ABI 7500 fast real-time PCR systems (Applied Biosystems) at 95 °C for 10 min and 40 cycles at 95 °C for 15 s and 60 °C for 60 s.

After validating the Ct mean of the housekeeping genes, U6snRNA, miR-16, and miR-1228, as reference miRNAs, the relative levels of each specific miRNA was calculated using the equation 2^-^∆Ct, where ∆Ct = mean Ct_miRNA_-mean Ct_(miR-U6,16&1228)_, and Ct = threshold cycle.

### 4.8. Statistical Analysis

The differences in circulating miRNA levels and the ROC curves were evaluated using IBM SPSS Statistics V22.0 (SPSS, Chicago, IL). Kruskal-Wallis nonparametric test was used to perform a statistical analysis of serum miRNA levels; for post hoc pairwise comparison, U Mann-Whitney test was used. The median expression level of each miRNA between different groups was compared (cancer vs. HDs).

For the multivariate analysis, Fisher linear discriminant analysis was employed. In order to establish the feasibility of this kind of analysis, assumptions of the model in terms of inequality of covariances and variances were checked with M-Box and Lambda of Wilks tests, respectively. The obtained discriminant function was used to classify the samples as tumor or normal. Performances of univariate and multivariate analysis were studied with the AUC of ROC curves.

### 4.9. miRNA Target-Interactions Analysis

In this analysis, miRNet tool was used (url: https://www.mirnet.ca/miRNet/home.xhtml) [41].

Briefly, this resource implements algorithms for differential expression and mining in miRNA databases in order to create a graph representing the relationships between miRNAs and its targets. Networks can be manually amplified and pruned. Finally, it implements hypergeometric tests to perform a gene set enrichment analysis with KEGG pathways [42].

## 5. Conclusions

In conclusion, with this study, we have defined a series of circulating miRNAs that define different tumor types with a very high diagnostic performance. In addition, these miRNAs would constitute the basis of a multianalyte blood test through the SAPHELY detection system; however, their clinical utility should be demonstrated in a prospective study in the context of a CSP.

## Figures and Tables

**Figure 1 ijms-21-02783-f001:**
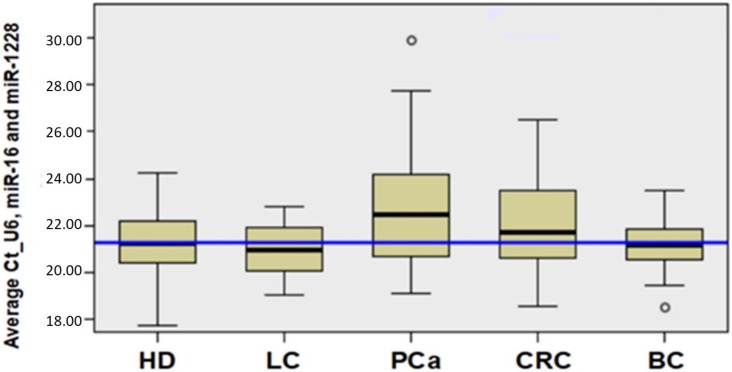
Selection of housekeeping genes: According to the null hypothesis, the median value of Ct of housekeeping genes has nonsignificant differences (*p* = 0.165), and the mean of Ct of U6snRNA, miR-16, and miR-1228 was used as a reference. Test used: nonparametric test for independent samples. The median value for each group is defined by the bold, black line within the box.

**Figure 2 ijms-21-02783-f002:**
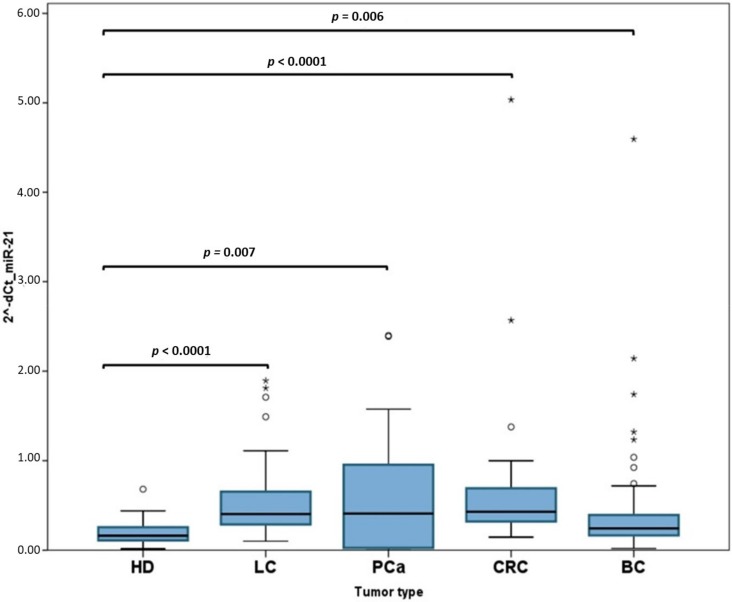
Levels of miR-21 expression in different tumor groups: All cancer groups show a significantly higher expression of miR-21 compared with healthy donors (HDs). It should be noted that prostate cancer (PCa) cases report the highest dispersion for miR-21 expression.

**Figure 3 ijms-21-02783-f003:**
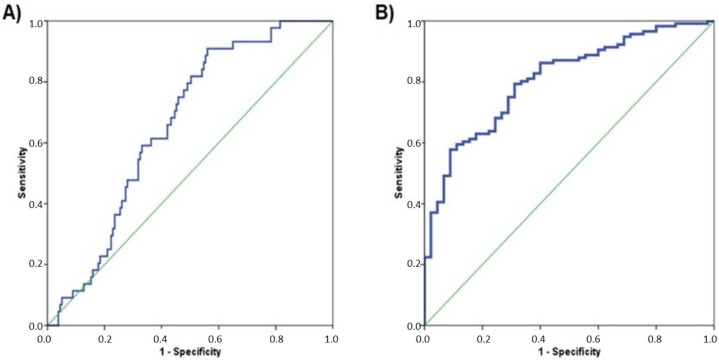
Receiver operating characteristic (ROC) curves for miR-21. (**A**) ROC curve including the patients for the four cancer groups: Area Under the Curve (AUC) = 0.675 (95% CI: 0.578–0.736; *p* = 0.001). (**B**) ROC curve excluding the cohort of PCa patients: AUC = 0.808 (95% CI: 0.738–0.878; *p* < 0.0001). ROC curve calculated is represented with blue line and green line is used as a reference.

**Figure 4 ijms-21-02783-f004:**
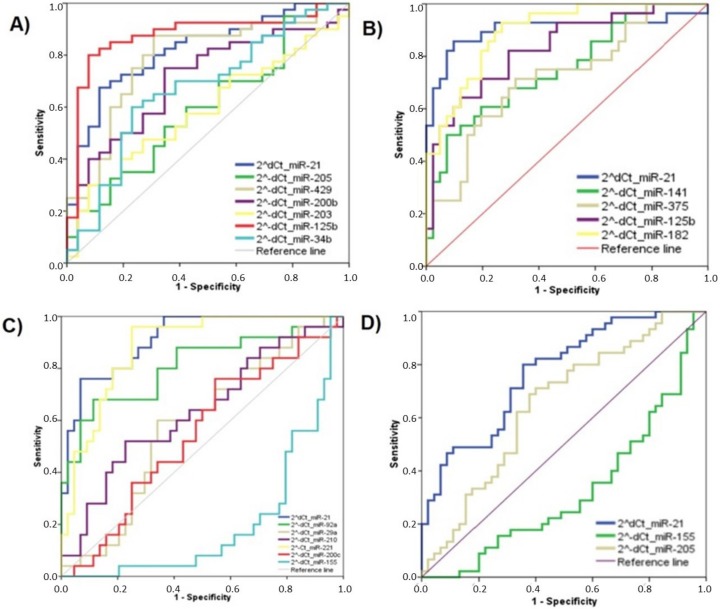
ROC curves of the selected circulating miRNAs selected for each tumor type: (**A**) LC for miR-21, miR-205, miR-429, miR-200b, miR-203, miR-125b, and miR-34b; (**B**) PCa for miR-21, miR-141, miR-375, miR-125b, and miR-182; (**C**) CRC, for miR-21, miR-92a, miR-29a, miR-210, miR-221, miR-200c and miR-155; and (**D**) BC, for miR-21, miR-155, and miR-205.

**Figure 5 ijms-21-02783-f005:**
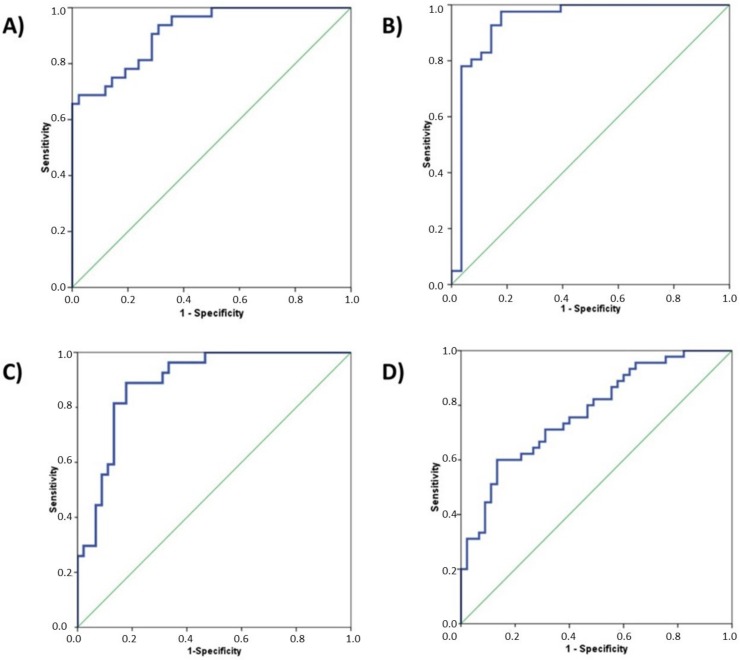
ROC curves of the combination of circulating miRNAs for each tumor type: (**A**) LC (miR-21, miR-429, miR-200b, and miR-125b); (**B**) PCa (miR-21, miR-141, miR-375, miR-125b, and miR-182); (**C**) CRC (miR-29a, miR-210, miR-200c, and miR-155); and (**D**) BC (miR-21 and miR-205). ROC curve calculated is represented with blue line and green line is used as a reference.

**Table 1 ijms-21-02783-t001:** Results of the ROC curve analysis: Summary of the results for each of the selected miRNAs according to tumor types.

Tumor Type	miRNA	AUC	*p*-Value	95% CI	
LC	miR-21	0.821	0.000012	0.720	0.923
	miR-205	0.597	0.185027	0.458	0.737
	miR-429	0.786	0.000097	0.670	0.902
	miR-200b	0.705	0.005186	0.577	0.833
	miR-203	0.587	0.237571	0.449	0.724
	miR-125b	0.877	2.6858 × 10^−7^	0.785	0.969
	miR-34	0.669	0.020907	0.534	0.805
PCa	miR-21	0.902	1.7668 × 10^−8^	0.809	0.994
	miR-141	0.753	0.000394	0.635	0.870
	miR-375	0.720	0.002074	0.596	0.843
	miR-125b	0.832	0.000003	0.734	0.930
	miR-182	0.895	2.8917 × 10^−8^	0.824	0.967
CRC	miR-21	0.913	1.4479 × 10^−8^	0.848	0.978
	miR-92a	0.809	0.000022	0.694	0.924
	miR-29a	0.579	0.277441	0.441	0.717
	miR-210	0.633	0.068360	0.494	0.772
	miR-221	0.882	1.5783 × 10^−7^	0.804	0.960
	miR-200c	0.547	0.516238	0.407	0.687
	miR-155	0.202	0.000042	0.095	0.308
BC	miR-21	0.771	0.000010	0.676	0.866
	miR-155	0.324	0.004121	0.213	0.436
	miR-205	0.649	0.014643	0.535	0.764
Lung Cancer (LC); Prostate Cancer (PCa); Colorectal Cancer (CRC); Breast Cancer (BC)					

**Table 2 ijms-21-02783-t002:** Performance of a tetra-panel of circulating miRNAs in identifying cancer patients.

Tumor Type	miRNAs	AUC	*p*-Value	95% CI	
LC	miR-21 miR-429 miR-200b miR-125b	0.914	1.2217 × 10^−9^	0.853459	0.975410
PCa	miR-21 miR-141 miR-375 miR-125b miR-182	0.937	8.5464 × 10^−10^	0.866459	1.000000
CRC	miR-21 miR-92a miR-221	0.891	3.1869 × 10^-−8^	0.818095	0.964621
BC	miR-21 miR-205	0.773	0.000008	0.677906	0.867773

**Table 3 ijms-21-02783-t003:** Clinical and histopathological characteristics of patients.

Tumor Type	*n*	Median Age (years) (Range)	Gender (M, Males; F, Females) (%)	Diagnosis (*n*)	Stage (*n*)
HDs	45	56.5 (50–73)	M, 29 (64%) F, 16 (36%)	-	-
CRC	27	70.5 (43–86)	M, 18 (67%) F, 9 (33%)	ADC (24) SRCC (1) Tubular ADC (1) MC (1)	I-IIA (15) IIIB-IV (12)
BC	45	57 (36–87)	F, 45 (100%)	IDC (35) ILC (5) DC in situ (4) PC (1)	0 (8) IA (13) IIA-IIB (22) IIIA-IIIB (2)
PCa	40	66 (46–78)	M, 40 (100%)	ADC (40)	cT1c (35) cT2a (5)
LC	44	61 (51–82)	M, 31 (70%) F, 13 (30%)	ADC (33) SCC (8) SC (1) NC (2)	IA-IB (28) IIA-IIB (7) IIIA-IV (9)

Adenocarcinoma: ADC; Signet ring cell carcinoma: SRCC; Tubular adenocarcinoma with high-grade dysplasia: Tubular ADC; Mucinous carcinoma: MC; Invasive ductal carcinoma: IDC; Invasive lobular carcinoma: ILC; Ductal carcinoma in situ: DC in situ; Papillary carcinoma: PC; Squamous cells carcinoma: SCC; Sarcomatoid carcinoma: SC; Neuroendocrine carcinoma: NC.

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
