# Peer review of "A Tetra-Panel of Serum Circulating miRNAs for the Diagnosis of the Four Most Prevalent Tumor Types"

_ijms, 2020, doi:10.3390/ijms21082783_

Round 1

Reviewer 1 Report

An excellent article by Pastor-Navarro et al. Micr RNAs have been of great interest as screening tools for cancer. the manuscript delves into a screening test using a set of predefined miRNAs for four different conditions and assessing their utility based on ROC. 

Major comments

  1. It is well known that miRNAs could be used as screening tools. One interesting aspect here is the varying stage at which cancer is diagnosed. The concentration of circulating miRNAs may vary depending upon the stage at which cancer is diagnosed. A stage 1 cancer may have a different concentration than one in metastatic stage. I do not see that aspect being considered either in analyses or in discussion. It may be a good idea to clarify this. If the number is too small for the patients to be stratified based on the stage of cancer and miRNA to be re-analyzed, please clarify that aspect.

Minor comments

  1. If the patients had any other markers such as PSA for prostatic cancer, was there any correlation between such markers and miRNA.
  2. Figure 1 line 88: English language may be checked and corrected.

Reviewer 2 Report

The manuscript is well-written with good analysis of MiRNA presented in different types of cancer. The authors can include cartoon in discussion of their findings with application of the findings to clinical research or cancer signaling. The authors can extend discussions on differences of miRNA in each type of cancer with potential explanations depending on cancer-specific. Signaling. 

Round 2

Reviewer 1 Report

The authors have addressed my comments satisfactorily